# Transformation Mechanisms of Chemical Ingredients in Steaming Process of *Gastrodia elata* Blume

**DOI:** 10.3390/molecules24173159

**Published:** 2019-08-30

**Authors:** Yun Li, Xiao-Qian Liu, Shan-Shan Liu, Da-hui Liu, Xiao Wang, Zhi-Min Wang

**Affiliations:** 1National Engineering Laboratory for Quality Control Technology of Chinese Herbal Medicines, Institute of Chinese Materia Medica, China Academy of Chinese Medical Sciences, Beijing 100700, China; 2Key Laboratory of TCM Quality Control Technology, Shandong Analysis and Test Center, Qilu University of Technology (Shandong Academy of Sciences), Jinan 250014, China; 3College of Pharmacy, Hubei University of Chinese Medicine, Wuhan 430065, China

**Keywords:** *Gastrodia elata*, steaming, gastrodin, parishin A, hydrolysis

## Abstract

To explore the transformation mechanisms of free gastrodin and combined gastrodin before and after steaming of *Gastrodia elata* (*G. elata*), a fresh *G. elata* sample was processed by the traditional steaming method prescribed by Chinese Pharmacopoeia (2015 version), and HPLC-ESI-TOF/MS method was used to identify the chemical composition in steamed and fresh *G. elata*. Finally, 25 components were identified in *G. elata* based on the characteristic fragments of the compounds and the changes of the 25 components of fresh and steamed *G. elata* were compared by the relative content. Hydrolysis experiments and enzymatic hydrolysis experiments of 10 monomer compounds simulating the *G. elata* steaming process were carried out for the first time. As a result, hydrolysis experiments proved that free gastrodin or *p*-hydroxybenzyl alcohol could be obtained by breaking ester bond or ether bond during the steaming process of *G. elata*. Enzymatic experiments showed that steaming played an important role in the protection of gastrodin, confirming the hypothesis that steaming can promote the conversion of chemical constituents of *G. elata*—inhibiting enzymatic degradation. This experiment clarified the scientific mechanism of the traditional steaming method of *G. elata* and provided reference for how to apply *G. elata* decoction to some extent.

## 1. Introduction

Rhizome of *Gastrodia elata* Blume (RGE), “TianMa” in Chinese, is considered a valuable traditional Chinese medicine, and has been used for the treatment of convulsions, epilepsy, tetanus, vertigo and paralysis [1,2,3] for thousands of years. Modern pharmacological studies have shown that RGE extract has neuroprotective, antioxidant, anti-anxiety, anti-depressant, sedative, anti-inflammatory and anti-angiogenic effects [4,5,6,7,8,9,10]. Gastrodin (GAS) and *p*-hydroxybenzyl alcohol (HBA) have been considered to be the active ingredients for a long time, due to their significant pharmacological activities [11,12,13,14,15]. Therefore, they were listed as the quality evaluation indicators of RGE in the Chinese Pharmacopoeia all along. However, researchers have made new important discoveries in recent years. Liu et al. [16] found that parishin C (PC) and parishin A (PA) could improve scopolamine damage in mice, affecting spatial learning and memory capacity significantly. The activity of PC was better than that of PA, but both were better than that of GAS. Shin et al. [17] found that PC could significantly improve the abnormal behavior of phencyclidine-induced schizophrenia in mice. Lin et al. [18] found that PC significantly ameliorated long-term potentiation (LTP) injury induced by amyloid β1-42 (Aβ1-42). In addition, it was found that PA had anti-aging effects [19], while parishin B (PB) had anti-asthma effects [20]. These findings suggested that parishins played a very important role in pharmacodynamics of RGE and their contents were much higher than that of GAS [21,22]. However, different drying methods had a great impact on the above chemicals of RGE. This led some researchers to propose that the majority of GAS were the product of the processing and steaming process methods of RGE, leading to significant differences in the contents of GAS [23,24]. In this paper, we studied the chemical transformation mechanisms of the traditional steaming method of RGE, to provide guidance for its reasonable use and production.

Steaming is a traditional technology for processing RGE in its producing areas and is important for its quality assurance [25,26,27]. Through researching records of herbal medicines in past dynasties, it can be found that RGE was first directly dried, then sliced and slightly boiled—or cooked in boiling water—after Song Dynasty, instead of being sliced raw. It can be seen that the steaming process of RGE has a long history of more than 1000 years. It is not only consistent with the lawful methods in the Chinese Pharmacopoeia, but also with the traditional processing methods in producing areas. The formation of the steaming process technology for RGE is the result of repeated practice and clinical validation by traditional Chinese medicine physicians in past dynasties. Based on the analysis of a large number of ancient and modern literatures and previous research works, we put forward the hypothesis that steaming can promote the transformation of chemical components and inhibit enzymatic degradation of RGE. That is, steaming can promote the chemical transformation of components such as parishins and, at the same time, it can inhibit the degradation of GAS by enzymolysis. Finally, a new dose-effect relationship was formed in RGE decoction. Based on this hypothesis, samples collected from producing areas were used as the research material for the experiment and modern analytical techniques were used to study the dynamic changes of chemical components and the mechanism of transformation of active ingredients in the RGE steaming process.

## 2. Results

### 2.1. Optimization of UHPLC/Q-TOF-MS/MS Conditions

In order to explore the mechanism of chemical constituents of RGE before and after steaming, it is important to optimize the chromatographic conditions, including suitable mobile phase system, gradient elution system, and detection wavelength, which are important for obtaining separation of various active ingredients. The UV wavelength at 270 nm was chosen. The mobile phase system was acetonitrile-water containing 0.1% phosphoric acid system. Optimized gradient elution is described in Section 3.4. The sample was dissolved in 80% methanol and ultrasound, for 30 min. The appropriate analytical temperature was set at 25 °C and the flow rate was 1.0 mL/min. The RGE-fresh (A) and RGE-steamed products (B) were analyzed under the conditions of Section 3.4 to obtain an HPLC chromatogram (Figure 1), which produced sharp peaks and peak shape symmetry, good separation, and prevention of peak tailing. Finally, 25 compounds were identified and labeled in Figure 1.

### 2.2. Structural Elucidation of Parishins by LC/MS

Peak 17 in positive ion mode, the fragment peak ions *m*/*z* 107.0492, 767.1714, 1019.3028, in which *m*/*z* 107.0492 is the fragment peak of benzyl alcohol, and the fragment ion *m*/*z* 767.1714 was [M − 269.1025 + K]^+^, fragment ion *m*/*z* 1019.3028 was [M + Na]^+^. So, it was judged that the compound had a molecular formula of C_45_H_56_O_25_, and the retention time was the same after comparison with the standard compound. Finally, it was confirmed that the compound was PA. Similar to peak 17, peaks 13 and 15 were identified as PB and PC, respectively. The fragment peak ions were *m*/*z* 107.0495, *m*/*z* 499.0766, *m*/*z* 751.2088, in which *m*/*z* 107.0495 was benzyl alcohol fragment, fragment peak ion *m*/*z* 499.0766 was [M − 269.1025 + Na]^+^ and *m*/*z* 751.2088 was [M + Na]^+^. The retention times were the same after comparison with the standards and it was determined that peaks 13 and 15 were PB and PC, respectively. Comparing to the published report [28], it was judged that peak 18 was parishin L (PL) and the fragment *m*/*z* 1049.3098 was [M + Na]^+^. Peak 9 was parishin G (PG) and the fragment *m*/*z* 483.1125 was [M + Na]^+^. Peak 10 was PG and the fragment *m*/*z* 483.1139 was [M + Na]^+^. Peak 12 was parishin J (PJ) and the fragment *m*/*z* 497.1252 was [M + Na]^+^. Peak 16 was parishin K (PK) and the fragment *m*/*z* 765.2213 was [M + Na]^+^. Peak 25 was parishin D (PD) and the fragment *m*/*z* 427.0998 was [M + Na]^+^. According to the mass spectrometry data (as shown in Table 1), a total of 16 kinds of parishin compounds and 9 other types of compounds were identified, and their structures are shown in Figure 2.

### 2.3. Effects of Steaming Process on 25 Compounds in RGE

To study the chemical conversion during the steaming process of RGE, 25 compounds were identified and relatively quantified. The results showed that chemical conversion occurred under steaming. The peak areas of 25 compounds were recorded as 1, and the relative contents of 25 compounds in the steamed products are shown in Table 2.

The holistic chemical profiles of fresh and steamed products were systematically compared by qualitative and quantitative analysis. Compared to fresh RGE, the relative peak areas of PA (17), parishin R (PR, 23), parishin T (PT, 21/22), parishin U (PU, 21/22), PL (18), parishin E (PE, 9), PJ (12), parishin H (PH, 14), parishin M (PM, 14), PK (16), parishin W (PW, 20), parishin S (PS, 23), PD (25), gastrodioside (19), 4,4’-dihydroxydibenzyl ether (24) decreased rapidly after steaming for 15 min. Meanwhile, GAS (6), HBA (8), PG (10), parishin B (PB, 13), and PC (15) increased gradually during steaming. This phenomenon indicated that higher molecular weights of parishins, gastrodioside and 4,4’-dihydroxydibenzyl ether degraded to their smaller molecular compounds, including GAS (6), HBA and other parishin compounds. We speculated that parishins, gastrodioside and 4,4’-dihydroxydibenzyl ether could be hydrolyzed under the steaming and acidic conditions and then influence the content of related compounds. Parishins of higher molecular weight are major components affecting the content of GAS (6) and HBA, which are the quality evaluation index of RGE in the Chinese Pharmacopoeia [1]. Many studies have revealed that parishins in RGE enhanced its bioactivities. As discussed above, the contents of GAS (6), HBA and parishins were unstable during the steaming process, indicating that the steaming time and temperature were important parameters of the processing procedure. Due to the chemical complexity of parishins, multi-components should be monitored for critically standardizing the steaming conditions and controlling the quality of RGE during the steaming process. The multi-components quantitative assessment could ensure the therapeutic effects of fresh and steamed products.

### 2.4. Summary of Steaming-induced Parishins Conversion in RGE

To study the chemical conversion during the RGE steaming process, the parishins were identified and relatively quantified. The results demonstrated that the chemical compositions transformation occurred under steaming. The transformation pathways of parishins are summarized in Figure 3. The compounds were all composed of citric acid and different amounts of GAS (6) or HBA. The protons on citric acid were replaced by R_1_, R_2_, R_3_, and the substituent groups included S_1_, S_2_ and S_3_. The hydrolysis pathway of parishins started from PA and lost different substituent groups in the direction of each arrow to form each compound. The characteristic transformation mechanisms detected were discussed.

PA **(**17) as original compounds transformed to PB (13), PC (15), PE (9) and PG (10) by the hydrolysis of ester bond and loss of different amounts of GAS fragments, leading the relative peak areas declined. PR (23), PT (21/22), PU (21/22) and PL (18) could transform to PB (13), PC (15), PE (9) and PG (10) identically. PK (16) could transform to PJ (12) by losing -OCH3. The contents of PH (14), PM (14) decreased after steaming because its special fragment S2 could not be converted from other compounds. Most of these transformation pathways could lose the fragment of S1 (the GAS fragment), so GAS (6) was a product of the steaming process. In the published reports, the transformation mechanisms and pathways of several parishins had been described during the steaming of fresh RGE [29]. The previous results partially agreed with our findings in related transformation mechanisms.

### 2.5. Experimental Verification of Hydrolysis of Monomeric Compounds

In view of the large difference among the contents of parishins, GAS, HBA, and other compounds in the traditional steaming process of RGE, it was speculated that parishins could be transformed to small molecular weight compounds, and a new dose-effect relationship was formed in the steamed product. In order to verify the hypothesis, 10 monomer compounds were selected to simulate the steaming process of RGE. LC/MS analysis of the hydrolyzed compounds represented by PA were carried out and six hydrolysates were identified.

#### 2.5.1. Structural Elucidation of Parishin A after Hydrolysis by LC/MS

The hydrolysis experiments were carried out and six hydrolysates (Table 3) of PA were analyzed and identified by LC/MS (Figure 4). The different amounts of GAS fragments in PE, PG, PB, and PC were obtained by thermal hydrolysis of PA under acidic conditions (Figure 5).

#### 2.5.2. Results of 10 Compounds after Hydrolysis

The previous study showed that PA could be hydrolyzed to GAS and HBA [30]. Our experiment verified for the first time that parishins could be hydrolyzed in the steaming process using monomeric compounds. Of the 25 compounds identified, we selected 10 compounds, including five parishins, two oxygen ethers, GAS, HBA and *S*-(4-hydroxybenzyl)glutathione to simulate the steaming process. The juice of fresh RGE was measured before hydrolysis, and the pH was 5.25. Therefore, the hydrolysis condition of 10 monomer compounds was steaming for 15 min after being dissolved in acidic aqueous solution (pH 5.25). Under the condition, the sample was hydrolyzed and further analyzed by comparing the chromatograms before and after hydrolysis (Figure 3). The experimental results showed that under the current conditions, PA could be hydrolyzed to GAS, HBA, PE, PG, PB, and PC (Figure 4). Compounds of PJ, PB, PC, PK, gastrodioside and 4,4’-dihydroxydibenzyl ether were hydrolyzed after simulating the steaming process and the HPLC chromatograms are shown in Figure 6. PJ could be hydrolyzed to GAS and HBA. PB could be hydrolyzed to GAS, HBA, PE, and PG. PC could be hydrolyzed to GAS, HBA, and PE. PK could be hydrolyzed to GAS and PJ. Gastrodioside could be hydrolyzed to GAS and HBA. 4,4’-dihydroxydibenzyl ether (24) could be hydrolyzed to HBA (8) and HBA was major produce of compound 24. However, GAS, HBA, and *S*-(4-hydroxybenzyl)glutathione were still stable after hydrolysis (Figure 7).

It was known to us that GAS and HBA were contained as the structural parts of parishins, gastrodioside, and 4,4’-dihydroxydibenzyl ether. The above compounds were hydrolyzed under acidic conditions by breaking the ester bond or benzyl ether bond to obtain GAS and HBS, and PG, PB, PG were obtained by thermal hydrolysis of PA under acidic conditions. On the contrary, GAS, HBA and *S*-(4-hydroxybenzyl)glutathione were not hydrolyzed under acidic conditions. Because the positive charge of hydrogen proton can be dispersed by the phenyl, which makes it difficult to break the C-O bond. HBA and *S*-(4hydroxybenzyl)glutathione contains chemical bonds that were easily hydrolyzed by acid. Therefore, GAS, HBA and *S*-(4-hydroxybenzyl)glutathione remain unchanged after simulating the steaming process under acidic conditions. The above experiments verified the inferences in the chemical composition changes of RGE in the steaming process.

Compounds of GAS, HBA, *S*-(4-hydroxybenzyl)glutathione were still stable after hydrolysis and the HPLC is shown in Figure 7.

### 2.6. Enzymatic Hydrolysis of GAS

There have been many reports that GAS can be enzymatically degraded into HBA in fresh RGE, but no research has proved that. In this experiment, GAS was used to simulate the steaming process with β-d-glucosidase. The results showed that GAS could be degraded into HBA in small amounts (Figure 8 A). At the same time, the comparison was carried out by incubating at 35 °C for 2 h. Most of the GAS was degraded into HBA (Figure 8 B). Therefore, the experiment demonstrated that the steaming process could inactivate β-d-glucosidase and inhibit the enzymatic hydrolysis of GAS.

## 3. Materials and Methods

### 3.1. Plant Material

The fresh rhizomes of *Gastrodia elata* were collected at the plantation base of Xiaocao Ba (27°75′N, 104°21′E, 1834 m), Yunnan province, China, in December 2018. All samples were authenticated by Dr. Da-Hui Liu (Institute of Medicinal Plants, Yunnan Provincial Academy of Agricultural Sciences, China). A voucher sample (No. GE-201812) was deposited at Shandong Analysis and Test Center, Qilu University of Technology (Shandong Academy of Sciences), China.

### 3.2. Chemicals and Reagents

GAS (6), HBA (8), *S*-(4-hydroxybenzyl)glutathione (11), parishin J (12), parishin B (13), parishin C (15), parishin K (16), parishin A (17), gastrodioside (19) and 4,4’-dihydroxydibenzyl ether (24) were separated and purified from the dried RGE and identified by Key Laboratory of TCM Quality Control Technology, Shandong Analysis and Test Center with purity > 98% by HPLC [31,32]. β-d-Glucosidase was purchased from Shanghai Yuanye Biotechnology Co., Ltd. (Shanghai, China).

Acetonitrile of chromatographic grade were obtained by the United States Fisher Scientific (Waltham, MA, USA). Formic acid of chromatographic grade was purchased from Tianjin Kermel Chemical Reagent Co., Ltd. (Tianjin, China). Calibration solution of pH 6.86 and 9.18 were purchased from Shanghai Solarbio Life Science Co., Ltd. (Shanghai, China). Hydrochloric acid of analytical grade was purchased from Sinopharm Chemical Reagent Co., Ltd. (Beijing, China). Experimental water was obtained by Hangzhou Wahaha Group Co., Ltd. (Hangzhou, China).

### 3.3. Instrumentation

The chromatographic analysis was carried out on HPLC-DAD 1120 system (Agilent, Technologies, Santa Clara, CA, USA), and G6520A TOF/Q-TOF Mass Spectrometer with ESI Ion Source (Agilent, USA). Other instruments included MS205DU electronic analytical balance (METTLER TOLEDO Instruments Shanghai Co., Ltd., Shang, China) accurated to 0.01 mg, SB-3200DT ultrasonic cleaning machine (Ningbo Xinzhi Biotechnology Co., Ltd., Ningbo, China), BUCHI Rotary Evaporator (BUCHI, Flawil, Switzerland), and DELTA 320 PH meter (METTLER TOLEDO International Co., Ltd.).

### 3.4. Sample Preparation

Three fresh RGE of similar size dug from the same hole were labeled 1, 2, 3 as parallel samples. All samples were washed and sliced (1–2 mm). The water content was determined by drying method according to the Chinese Pharmacopoeia [1]. The average water content of the three parallel samples was determined as 72.57%. Sampling quantity of fresh products was converted according to the average water content measured and each was calculated according to dry products, about 2 g.

#### 3.4.1. Fresh Products

According to the above sampling method, three fresh RGE were sliced longitudinally and labeled 1, 2, 3 as parallel samples.

#### 3.4.2. Steaming Products

According to the above sampling method, three fresh RGE were sliced longitudinally, labeled 1, 2, 3 as parallel samples and then steamed. Proper amount of water was added to the steamer with cage drawer. The fresh slices of RGE were then put on the cage drawer after the water had boiled. The timing started after the round vapor. After 15 min, the RGE were taken out and cooled to room temperature.

The above-mentioned processed A and B of RGE samples were crushed and calculated according to dry products about 2 g. They were then immersed in 50 mL 80% ethanol by ultrasonic (180 W, 40 kHZ) extraction for 30 min. After filtration, 20 mL solution was concentrated to no alcohol taste. The residue was dissolved in acetonitrile–water (1:9) and transferred to 10 mL volumetric flask. All sample solutions were filtered through 0.22 µm filter membranes before being injected into HPLC system for analysis.

### 3.5. Chromatographic Procedures

Separation was performed on a Waters Symmetry^®^ C18 reverse phase column (250 mm × 4.6 mm, 5 μm) with column oven at 25 °C. The mobile phase consisted of 0.1% formic acid aqueous(A)–acetonitrile (B) at a flow rate of 1 mL/min. Due to the complexity of *G. elata* sample, the separation gradient mode was optimized as follows: 0–8 min, 1%–10% B; 8–22 min, 10%–20% B; 22–32 min, 20%–30% B; 32–45 min, 30%–90% B; 45–50 min, 90% B. The injection volume was set at 10 µL. The HPLC chromatograms of RGE-fresh (A) and RGE-steamed (B) samples was shown in Figure 1. Since the hydrolysis sample was simpler, and to save time, the separation gradient mode was optimized. The mobile phase consisted of 0.1% formic acid (A)–Methanol (C) and the gradient elution was carried out at 0–30 min, 5%–100% C. Other conditions were the same as the RGE extracted sample. The UV detection wavelength was set at 270 nm after investigation.

The ion source was an electrospray ionization source (ESI) using both the positive and negative ion modes. The full scan range was 100–2000 *m*/*z*, and the spray pressure was 40 psi; The drying gas flow rate was 12.0 L/min and the drying gas temperature was 350 °C. Under the optimized chromatographic conditions, the peaks were identified by high resolution electrospray ionization time-of-flight mass spectrometry. Based on the accurate molecular weight information of the obtained compound and the UV absorption information obtained by the DAD detector, the compounds were identified by reference to the relevant literature [28]. The results of the precise measurement of each compound were shown in Table 3.

### 3.6. Acid Hydrolysis Experiment

Proper amount of fresh RGE slices were ground in a mortar. The pH value of fresh RGE juice was 5.25 measured by pH meter. The above 10 monomer compounds were precisely weighed 1.0 mg, dissolved by 1.0 mL hydrochloric acid solution of pH 5.25, and placed in a vial. The sample solution of 1.0 mg/mL was obtained. The vials of 10 samples were steamed in the steamer simulating the steaming process of RGE. The timing started after the round vapor. After 15 min, the vials were taken out. After cooling to room temperature, the analysis was carried out under the condition of Section 3.5, PA was analyzed by HPLC-ESI-TOF/MS and the other nine compounds were analyzed by HPLC.

### 3.7. The β-d-glucosidase Enzymatic Hydrolysis Experiment

GAS (0.591 mg/mL) and β-d-glucosidase (1.24 mg/mL) were dissolved in hydrochloric acid solution of pH 5.25 in a vial. Two copies of the above samples were prepared. One was steamed on a steamer for 15 min, and the other was incubated in water bath at 35 °C for 2 h. A 10 µL aliquot was injected for HPLC analysis.

## 4. Conclusions

At the present study, 25 components in fresh and steamed *Gastrodia elata* rhizomes were identified by HPLC-ESI-TOF/MS, and the content change regularity was studied by HPLC. The content change of GAS, HBA, parishins, and others in the steaming process was verified by the simulated steaming hydrolysis experiments of 10 monomer compounds. It conformed to the theory that compounds containing ester bond and ether bond could be hydrolyzed under acidic and heated conditions. It was also verified that steaming could promote the transformation of chemical components of RGE and inhibit the enzymatic degradation. Based on above, the mechanism of the ancient steaming of RGE was explained which provided the basis for “inactive enzymes and protect glycosides” of RGE in production practice. Due to the chemical complexity in the variation of parishins, multi-components should be monitored for critically standardizing the steaming conditions and controlling the quality during the steaming process of RGE. Moreover, in order to control the quality of RGE, in-depth research should be carried out. Multi-components analysis, biological evaluation, pharmacological activity and other methods must be combined to evaluate the quality of *Gastrodia elata*.

## Figures and Tables

**Figure 1 molecules-24-03159-f001:**
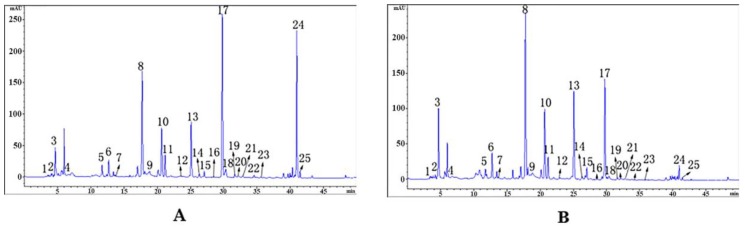
The HPLC chromatograms of RGE-fresh (**A**) and RGE-steamed (**B**) samples.

**Figure 2 molecules-24-03159-f002:**
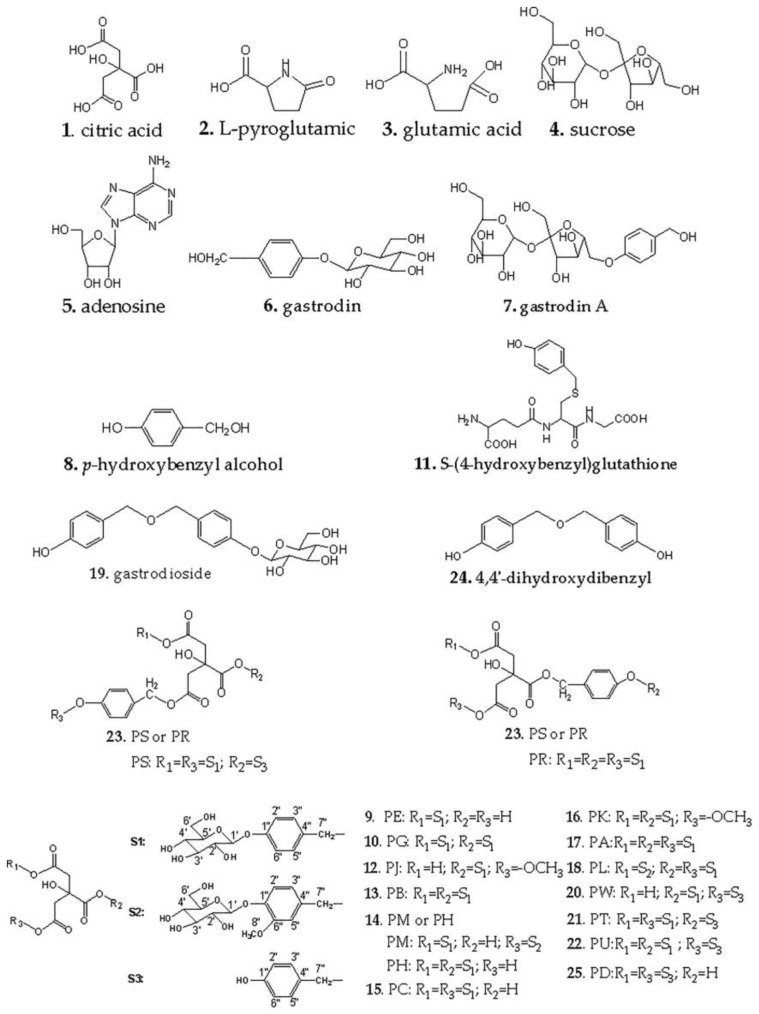
The structure of 25 compounds of RGE.

**Figure 3 molecules-24-03159-f003:**
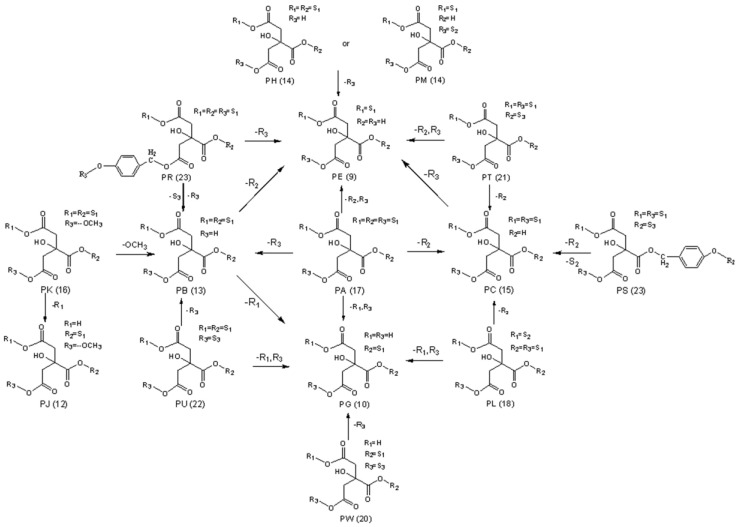
The hydrolysis pathways of parishins.

**Figure 4 molecules-24-03159-f004:**
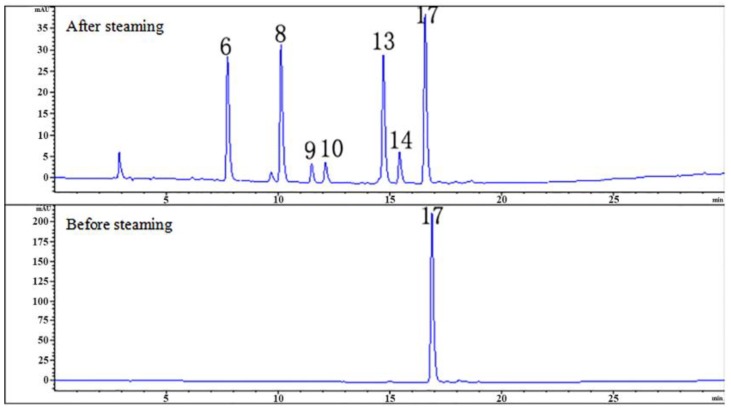
The HPLC chromatograms of PA before and after hydrolyzed.

**Figure 5 molecules-24-03159-f005:**
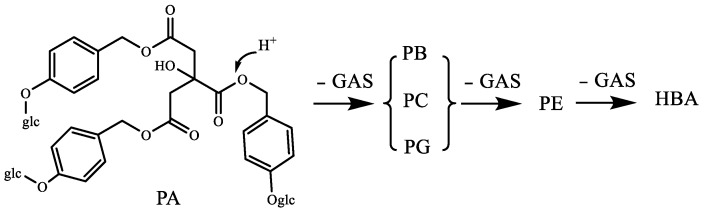
The hydrolysis pathways of parishin A.

**Figure 6 molecules-24-03159-f006:**
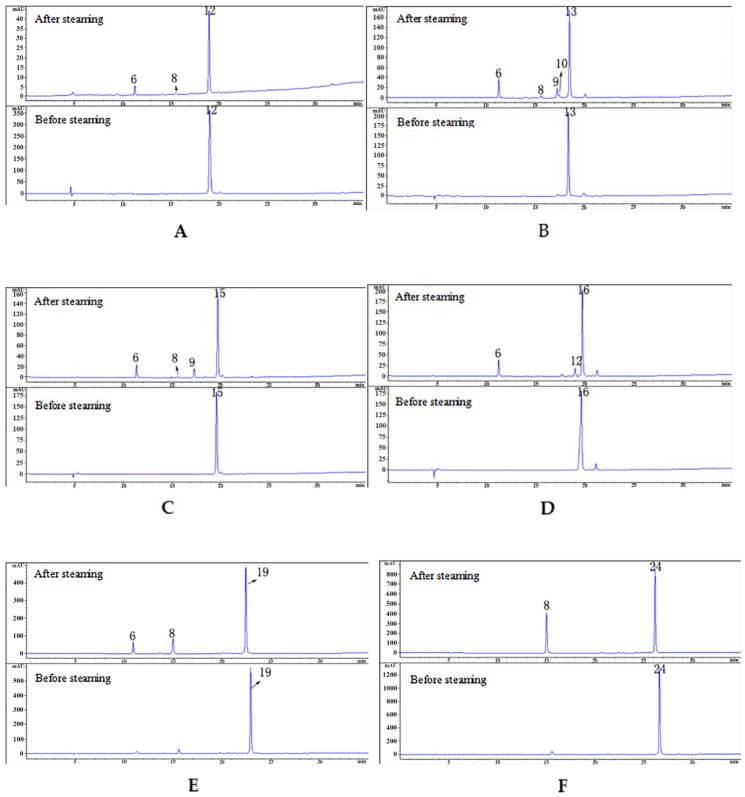
The HPLC chromatograms of compounds PJ (**A**), PB (**B**), PC (**C**), PK (**D**), gastrodioside (**E**), 4,4’-dihydroxydibenzyl ether (**F**) after hydrolysis.

**Figure 7 molecules-24-03159-f007:**
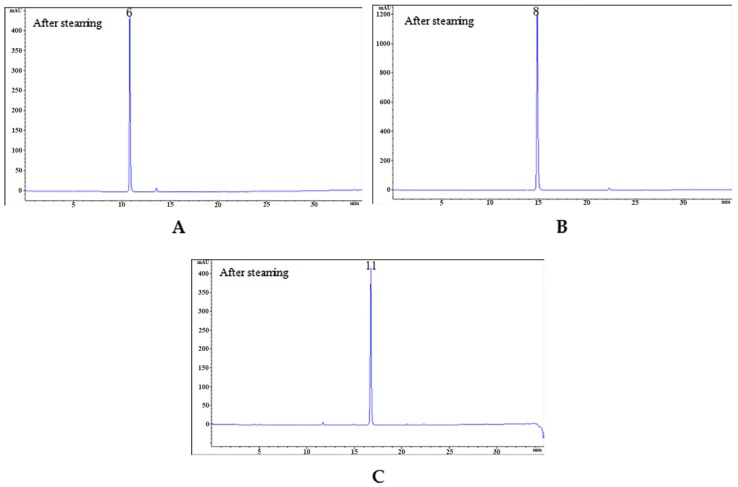
The HPLC chromatograms of GAS (**A**), HBA (**B**), *S*-(4-hydroxybenzyl)glutathione (**C**) after hydrolysis.

**Figure 8 molecules-24-03159-f008:**
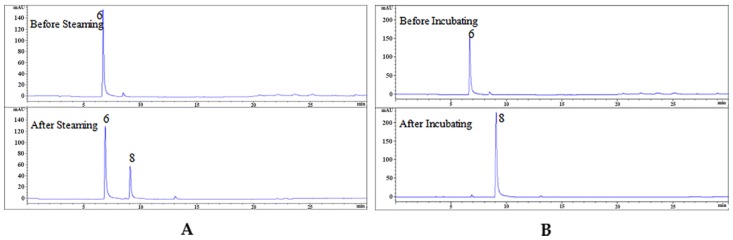
The HPLC chromatograms of GAS before and after enzymatic hydrolyzed steaming sample (**A**); incubating sample (**B**).

**Table 1 molecules-24-03159-t001:** ESI-TOF/MS accurate mass measurements of 25 compounds of RGE.

Peak No.	Retention Time/min	Compound	Molecular Formula	M	Fragment Peaks
1	3.53	citric acid	C_6_H_8_O_7_	192.0270	214.9171 [M + Na]^+^
2	4.06	l-pyroglutamic acid	C_5_H_7_NO_3_	129.0426	130.0490 [M + H]^+^
3	4.64	glutamic acid	C_5_H_9_NO_4_	147.0532	148.0608 [M + H]^+^
4	5.12	sucrose	C_12_H_22_O_11_	342.1162	365.1074 [M + Na]^+^
5	11.71	adenosine	C_10_H_13_N_5_O_4_	267.2413	268.1044 [M + H]^+^
6	12.69	gastrodin	C_13_H_18_O_7_	286.1052	309.0922 [M + Na]^+^
7	13.41	gastrodin A	C_19_H_28_O_12_	448.1581	471.1473 [M + Na]^+^
8	17.72	*p*-hydroxybenzyl alcohol	C_7_H_8_O_2_	124.0524	125.0507 [M + H]^+^
9	18.85	parishin E	C_19_H_24_O_13_	460.1217	483.1125 [M + Na]^+^
10	20.65	parishin G	C_19_H_24_O_13_	460.1217	483.1139 [M + Na]^+^
11	21.17	*S*-(4-hydroxybenzyl)glutathion	C_17_H_23_N_3_O_7_S	413.4454	414.1410 [M + H]^+^
12	23.54	parishin J	C_20_H_26_O_13_	474.4126	497.1252 [M + Na]^+^
13	25.10	parishin B	C_32_H_40_O_19_	728.2164	751.2087 [M + Na]^+^
14	26.79	parishin H/M	C_33_H_42_O_20_	758.2269	757.2201 [M − H]^−^
15	27.05	parishin C	C_32_H_40_O_19_	728.2164	751.2060 [M + Na]^+^
16	28.46	parishin K	C_33_H_42_O_19_	742.6752	765.2213 [M + Na]^+^
17	29.71	parishin A	C_45_H_56_O_25_	996.9111	1019.3028 [M + Na]^+^
18	30.29	parishin L	C_46_H_58_O_26_	1026.3216	1049.3098 [M + Na]^+^
19	31.46	gastrodioside	C_20_H_24_O_8_	392.3998	391.1414 [M − H]^−^
20	32.25	parishin W	C_26_H_30_O_14_	566.1636	589.1519 [M + Na]^+^
21	32.82	parishinT/U	C_39_H_46_O_20_	834.2583	833.2524 [M − H]^−^
22	34.54	parishin T/U	C_39_H_46_O_20_	834.2583	833.2522 [M − H]^−^
23	35.56	parishin R/S	C_52_H_62_O_26_	1102.3529	1101.3611 [M − H]^−^
24	40.80	4,4’-dihydroxydibenzyl ether	C_14_H_14_O_3_	230.0937	229.0873 [M − H]^−^
25	41.47	parishin D	C_20_H_20_O_9_	404.1107	427.0998 [M + Na]^+^

**Table 2 molecules-24-03159-t002:** The relative contents of 25 compounds in RGE.

Peak No.	Retention Time/min	Compound	Fresh Product (A)	Steamed Product (B)
1	3.53	citric acid	1	1.2599
2	4.06	l-pyroglutamic acid	1	0.9366
3	4.64	glutamic acid	1	1.9929
4	5.12	sucrose	1	-
5	11.71	adenosine	1	0.7680
6	12.69	gastrodin	1	1.3630
7	13.41	gastrodin A	1	0.5475
8	17.72	*p*-hydroxybenzyl alcohol	1	1.3769
9	18.85	parishin E	1	0.8431
10	20.65	parishin G	1	1.2545
11	21.17	*S*-(4-hydroxybenzyl)glutathione	1	0.8722
12	23.54	parishin J	1	0.3504
13	25.10	parishin B	1	1.4318
14	26.79	parishin H/M	1	0.9091
15	27.05	parishin C	1	1.7347
16	28.46	parishinK	1	0.4400
17	29.71	parishin A	1	0.5532
18	30.29	parishin L	1	0.2824
19	31.46	gastrodioside	1	1.0220
20	32.25	parishin W	1	0.7028
21	32.82	parishin T/U	1	0.6596
22	34.54	parishin T/U	1	0.6336
23	35.56	parishin R/S	1	0.6495
24	40.80	4,4’-dihydroxydibenzyl ether	1	0.0844
25	41.47	parishin D	1	0.3535

Note: “-” indicates that the content is not calculated.

**Table 3 molecules-24-03159-t003:** ESI-TOF/MS accurate mass measurements of seven compounds in samples.

Peak No.	Retention Time)/min	Compound	Molecule Formula	M	Fragment Peaks
6	7.73	gastrodin	C_13_H_18_O_7_	286.1052	309.0970 [M + Na]^+^
8	10.12	*p*-hydroxybenzyl alcohol	C_7_H_8_O_2_	124.0524	125.0507 [M + H]^+^
9	11.49	parishin E	C_19_H_24_O_13_	460.1269	483.1145 [M + Na]^+^
10	12.11	parishin G	C_19_H_24_O_13_	460.1269	483.1141 [M + Na]^+^
13	14.70	parishin B	C_32_H_40_O_19_	728.2163	751.2170 [M + Na]^+^
14	15.43	parishin C	C_32_H_40_O_19_	728.2163	751.2176 [M + Na]^+^
17	16.57	parishin A	C_45_H_56_O_25_	996.3110	997.3108 [M + H]^+^

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
