# Peer review of "Transformation Mechanisms of Chemical Ingredients in Steaming Process of Gastrodia elata Blume"

_molecules, 2019, doi:10.3390/molecules24173159_

Round 1

Reviewer 1 Report

The manuscript described the chemical constituents change by the steaming process of Gastrodia elata Rhizome. The chemical constituents in G. eleta rhizome before and after steaming process were identified by HPLC-ESI-TOF MS. The acidic and glucosidase hydrolysis of compounds were also used to simulate the steaming process degradation of compounds. However, in section 3.3 Sample preparation should describe more detail. Section 3.5 acid hydrolysis, why don’t use pH 5.25 buffer solution? Gastrodin contains ether bond linkage but in L191 stated it don’t contains? Ester bond are more easy hydrolysis in acidic condition than ether bond, it can be verified in this manuscript. The HBA (8) was major produce by hydrolysis of compound 24. It should be added in L 185. Authors only showed steaming processing after 15 min. How about more longer steaming time?

Other minor points should be corrected.

Figure 2, only partial chemical structures of compounds were provided and two compound 23 were present. One of compound 23 was same as compound 17. It should be corrected.

Figure 3, the compound numbers should provided. For example, PA (17).

Reviewer 2 Report

This paper reports analytic comparison of chemical composition between steamed and fresh Gastrodia elata and explore the transformation mechanisms of gastrodin and parishins before and after steaming. It clarified the scientific mechanism of the traditional steaming method of G. elata. Thus, this manuscript may be accepted for Molecules after minor revisions.

Provide the detail of steaming process in Materials and Methods section. Correct “s-(4-hydroxybenzyl)-giutathione” into “S-(4-hydroxybenzyl)glutathione” through the whole text.

Author Response

Response to reviewer’s comments:

Point 1: Provide the detail of steaming process in Materials and Methods section.

Response 1: Thanks very much for your affirmation and support to my research. I’m sorry for my unclear interpretation about sample preparation. More detail has been added in Page 11 - 12, section 3.3, lines 247-258.

 Point 2: Correct “s-(4-hydroxybenzyl)-giutathione” into “S-(4-hydroxybenzyl)glutathione” through the whole text.

Response 2: Thank you for your corrections and I’m sorry for my careless. We had corrected “s-(4-hydroxybenzyl)-giutathione” into “S-(4-hydroxybenzyl)glutathione” through the whole text.

Others:

Point 3: Many small errors were also corrected and marked in red in this modified manuscript.

Response 3: The wrong number has been corrected in line 174.

We are really grateful for the suggestions from reviewers. These suggestions were detailed and very valuable to improve our paper. We had revised this manuscript in light of the comments as carefully as we can. And I’m sorry for my poor English, we have made careful corrections in English thoroughly. If any question of this manuscript, please contact with us, we will revise it according to your comments as soon as possible.

Many thanks for your attention.

I am looking forward to your reply.

Round 2

Reviewer 1 Report

The compound names in Tables 1, 2, and 3 and in all manuscript (except begin of the sentence) should be not capitalized. 

In L. 254, The water content in fresh RGE should provided.

In L.265, authors mentioned two processed A and B. But in L.261-264, only process A described. 

Author Response

Dear Editor,

Thank you for your comments concerning our manuscript entitled "Transformation mechanisms of Chemical Ingredients in Steaming Process of Gastrodia elata Blume" (ID: molecules-564960). We greatly appreciate for your kindly comments on our manuscript. These comments have all been valuable and very helpful for revising and improving our manuscript. We have studied comments carefully and have made revisions which we hope meet with approval. The main corrections in the paper are in track changes mode and the point-by-point response to the reviewer’s comments are as follows. Thank you so much for your precious time spent on our manuscript.  

Response to reviewer’s comments:

Point 1: The compound names in Tables 1, 2, and 3 and in all manuscript (except begin of the sentence) should be not capitalized.

Response 1: Thank you for your valuable comments. The capital letters of compound names in Tables 1, 2, and 3 and in all manuscript (except begin of the sentence) have been modified.

Point 2: In L. 254, The water content in fresh RGE should provided.

Response 2: Thank you for your reminding. I'm sorry that I forgot to provide the water content of fresh RGE. The average water content of three parallels determined acccording to Chinese Pharmacopoeia was 72.57%, which has been added to L. 248 - 249.

Point 3: In L.265, authors mentioned two processed A and B. But in L.261-264, only process A described. 

Response 3: Thank you for your comments and I’m sorry for my unclear descriptions. In L.261 - 264, the extraction methods of A and B samples was described in the modified manuscript. The preparation process of two samples was described in L.251 - 259. And the chromatographic procedures of A and B samples was describes in L.266.

Others:

Point 4: The descriptions of sentence in L.271 was inaccurate. The sentence of “The reference and the sample (S1) chromatogram was shown in Figure 1” have been modified into “The HPLC chromatograms of RGE-fresh (A) and RGE-steamed (B) samples was shown in Figure 1”.